# Progesterone Receptor Membrane Component (PGRMC)1 and PGRMC2 and Their Roles in Ovarian and Endometrial Cancer

**DOI:** 10.3390/cancers13235953

**Published:** 2021-11-26

**Authors:** John J. Peluso, James K. Pru

**Affiliations:** 1Department of Cell Biology, University of Connecticut Health Center, Farmington, CT 06030, USA; 2Department of Obstetrics and Gynecology, University of Connecticut Health Center, Farmington, CT 06030, USA; 3Department of Animal Science, Program in Reproductive Biology, University of Wyoming, Laramie, WY 82071, USA; jpru@uwyo.edu

**Keywords:** cancer, endometrium, ovary, PGRMC1, PGRMC2

## Abstract

**Simple Summary:**

Ovarian and endometrial cancers occur frequently and can be lethal. Recently, progesterone receptor membrane component (PGRMC) 1 and PGRMC2 have been shown to be highly expressed in these cancers and play important roles in promoting tumor growth and chemoresistance. This review will focus on the mechanisms through which these two PGRMC family members influence the growth of ovarian and endometrial cancers. The mechanisms highlighted in this review could also provide insights into the roles that these PGRMCs play in regulating the fate of other PGRMC-expressing cancers such as lung, breast, and pancreatic cancer.

**Abstract:**

Cancers of the female reproductive tract are both lethal and highly prevalent. For example, the five-year survival rate of women diagnosed with ovarian cancer is still less than 50%, and endometrial cancer is the fourth most common cancer in women with > 65,000 new cases in the United States in 2020. Among the many genes already established as key participants in ovarian and endometrial oncogenesis, progesterone receptor membrane component (PGRMC)1 and PGRMC2 have gained recent attention given that there is now solid correlative information supporting a role for at least PGRMC1 in enhancing tumor growth and chemoresistance. The expression of PGRMC1 is significantly increased in both ovarian and endometrial cancers, similar to that reported in other cancer types. Xenograft studies using human ovarian and endometrial cancer cell lines in immunocompromised mice demonstrate that reduced expression of PGRMC1 results in tumors that grow substantially slower. While the molecular underpinnings of PGRMCs’ mechanisms of action are not clearly established, it is known that PGRMCs regulate survival pathways that attenuate stress-induced cell death. The objective of this review is to provide an overview of what is known about the roles that PGRMC1 and PGRMC2 play in ovarian and endometrial cancers, particularly as related to the mechanisms through which they regulate mitosis, apoptosis, chemoresistance, and cell migration.

## 1. Introduction

As will be outlined in this review, cancers often develop within the ovary and uterus. Although the ovary and uterus have very different functions, the cancers that develop within these organs have several common characteristics. In fact, some findings suggest that ovarian epithelial cancers are derived from either endometrial or fallopian tube epithelium [1] and are sometimes collectively referred to as pelvic floor cancers [2]. Moreover, both cancers are hormonally regulated [3,4,5] and they express various progesterone (P4) receptors [1,3,4]. One characteristic that is not often appreciated is that they express members of the P4 receptor membrane component (PGRMC) family, most notably PGRMC1 and PGRMC2 (for review see [1]). PGRMC1 was the first family member to be identified. It was detected in the membrane fraction of liver cells based on its capacity to bind P4, and hence named P4 receptor membrane component 1 [6]. PGRMC2 was identified based on the similarity of its sequence [2]. However, more in-depth studies revealed that these PGRMC proteins were detected throughout the cell [4] and influenced the development of both ovarian [4] and endometrial cancer [3,4]. Given these common characteristics, this review will focus on these two PGRMC family members and how they influence the growth and development of ovarian and endometrial tumors. Although focusing on ovarian and endometrial cancer, we will also present data obtained from normal ovarian and uterine cells as well as cell lines in order to provide more detailed information related to the mechanisms of action of PGRMC1 and PGRMC2.

## 2. PGRMC1 and PGRMC2 in Ovarian and Endometrial Cancer

### 2.1. PGRMC1 and PGRMC2 as Mediators of P4′s Actions

Members of the PGRMC family are essential for female reproduction [7,8]. Within this protein family, PGRMC1 and PGRMC2 are the best characterized as they mediate some of the actions of P4, and PGRMC1 has been shown to harbor a high affinity P4 binding site [4,6,9]. Interestingly, the evolutionary origin of *PGRMC* genes predates the appearance of the classical sex steroid receptor family [10,11]. While sex steroid hormone receptors are thought to have evolved in early metazoans [12,13], the prototypical member of the *PGRMC* family was present in the common ancestor of opisthokonts and plants, possibly 1 billion years prior to metazoans [11]. To complicate matters further, cell membrane fractions with high PGRMC1 content were originally reported to have high binding affinities for P4, corticosterone, testosterone, and cortisol [6]. As such, it is now becoming increasingly clear that PGRMCs have both P4-dependent and P4-independent functions and that they may more broadly serve as promiscuous receptors for select sterols. This is supported by recent findings in breast cancer cell lines that different synthetic progestins cause many of the same cellular outcomes when signaling through the classical or nuclear progesterone receptor (PGR), but have seemingly distinct PGRMC1-dependent actions [14,15]. For example, medroxyprogesterone acetate, norethisterone, and dienogest, but not nomegestrol acetate, induce proliferation in MCF7 and T47D breast cancer cells by a mechanism that depends on phosphorylation of a casein kinase 2 domain within PGRMC1 [15,16,17]. Since *PGRMC* genes are evolutionarily conserved, it is not surprising that they are expressed in various cancers and influence the function of cancer cells as outlined throughout this Special Issue of *Cancers.*

### 2.2. Overview of the Morbidity and Mortality of Ovarian and Endometrial Cancers

Ovarian cancer is the eighth most common cancer, due in large part to its asymptomatic nature and it being diagnosed at an advanced stage. Ovarian cancers also have a tremendous metastatic potential that accounts for it being the fifth leading cause of cancer-related deaths among the female population [18]. Ovarian neoplasms are derived from various cellular sources with adenomas and adenocarcinomas being of epithelial origin in either the overlying ovarian surface epithelium or epithelial cells of the fallopian tube or uterus. Ovarian adenomas and adenocarcinomas have the highest mortality rate in women with ovarian cancer, whereas ovarian neoplasms derived from the ovarian stroma or germ cells have a much higher survival rate [19].

There are several factors that may lead to the development of ovarian cancer. A major contributing factor is a previous family history of the disease, which increases the risk of developing ovarian cancer by 3–7 fold. The most significant genetic risk factors are mutations in the BRCA1 and BRCA2 genes and Lynch’s syndrome [14,18,20,21]. Another major risk factor includes high lifetime ovulation numbers. As such, women who have an early menarche, do not have children, and reach menopause beyond the average age of 52 years are more likely to develop ovarian cancer. This indicates a link between the oxidative damage to ovarian tissues caused by ovulation and development of the disease [18,22]. Other environmental risk factors that predispose women to the development of ovarian cancer include obesity, smoking, alcohol consumption, diabetes, and dietary intake of acrylamide and endocrine disruptors, as well as various medications including certain forms of hormone replacement therapy [14,18,20,21,23,24]. Despite recent efforts to design better therapeutics based on these risk factors, the global incidence and mortality rates have not significantly changed over the last three decades. Early detection and diagnosis are still viewed as giving the best chance of surviving this disease.

Although ovarian cancer accounts for more cancer-related deaths, endometrial cancer is the most common gynecological malignancy with an estimated 65,620 new cases and 12,590 deaths in the United States in 2020 [25]. Both the incidence of new cases and deaths from endometrial cancer are increasing yearly. Endometrial cancers are classified as Type I or Type II. While Type II endometrial cancers tend to metastasize and are hormone-independent, Type I endometrial cancers are generally considered endocrine-dependent and commonly have mutations in PTEN, members of the PI3K/AKT pathway, and KRAS [26,27,28,29]. Prolonged unopposed exposure to estrogens through obesity (i.e., estrone), polycystic ovarian syndrome, progesterone-deficient hormone replacement therapy after the menopausal transition, and prolonged tamoxifen use are risk factors in developing Type I endometrial cancer [30,31,32,33,34,35].

As with many different types of cancers, PGRMC1 expression is elevated in ovarian and endometrial cancers [3,4]. The studies indicating that PGRMC1 and potentially PGRMC2 play important roles in the growth and development of these cancers and these issues are the focus of this review.

## 3. Expression and Subcellular Localization of PGRMC1 and PGRMC2 in Neoplastic Cells of the Ovary and Endometrium

### 3.1. PGRMC1 and PGRMC2 Subcellular Localizations

Studies using numerous cell types have shown that the subcellular localization of PGRMC1 is rather complex in that it is broadly present in the endoplasmic reticulum, Golgi apparatus, plasma membrane, mitochondria, and nucleus [2]. The diverse localization pattern of PGRMC1 is cell and context specific. It can also exist as a ≈ 25 kDa monomer or oligomers of 50 and 75 kDa or more [36] with the oligomers localizing to the nucleus [36]. In the nucleus of cultured human granulosa cells, PGRMC1 is not evenly distributed, but instead localizes in a punctate pattern [37]. PGR is also present in the nucleus at the precise focus that coincides with PGRMC1 and in fact in cultured human granulosa cells, nuclear PGR and PGRMC1 interact as revealed by in situ proximity ligation assay [37]. Neither the functional significance nor whether this interaction occurs in cancer cells has been determined. However, it seems likely that each of these cellular sites influences the function of PGRMC1, which may or may not be dependent on P4. How the subcellular sites of PGRMC1 influence its function will be discussed later in this review and in a review in this issue of *Cancers* by Dr. Valentina Lodde and associates.

Interestingly, PGRMC2 localizes mainly to the cytoplasm and under some conditions interacts with PGRMC1. Both PGRMC1 and PGRMC2 are heme binding proteins and each has multiple phosphorylation, acetylation, and sumoylation domains within their primary amino acid sequences. This suggests that each PGRMC protein is regulated through complex interactions with kinases/phosphatases, acetylases/deacetylases, and ubiquitin-like enzymes [2,10,38,39]. All of these factors likely contribute to the complexity of defining the function, diverse subcellular distribution, and mechanism of action of these two proteins.

### 3.2. Cellular Localization and Expression Levels within Neoplastic Ovary

The relationship between the expression profile of PGRMC1 and PGRMC2 and ovarian cancer is complex because expression and cellular location vary with the subtype of ovarian cancer and the stage of progression. Various “omic” approaches provide basic information with much of this information summarized in the Expression Atlas database (http://www.ebi.ac.uk/gxa, accessed on 15 September 2021). This database organizes data on gene and protein expression levels with respect to species, tissue, cell type, and disease state [40]. Searching this database (July 2021) revealed that both mRNA and protein levels of PGRMC1 and PGRMC2 are significantly elevated in ovarian tumors. Moreover, studies of 17 different cell lines derived from human ovarian tumors demonstrate that these cell lines also overexpress both PGRMC1 and PGRMC2 (http://www.ebi.ac.uk/gxa, accessed on 15 September 2021).

These transcriptomic and proteomic studies are important but do not provide information on the expression of PGRMC family members in various subtypes of ovarian cancer. There are four main subtypes of ovarian cancers: mucinous, endometroid, clear cell, and serous, with papillary serous cystadenocarcinomas being the most common malignant ovarian cancer. Immunohistochemical analysis detected PGRMC1 in each of these subtypes but at different levels of intensity. Compared to the non-cancerous ovary, PGRMC1 was present in greater abundance in clear cell carcinoma (Figure 1A), endometroid carcinoma (Figure 1B), and papillary serous cystadenocarcinomas with increasing staining intensity as these cancers progress from stage IIB (Figure 1C) to stage IIIC (Figure 1D) [41]. Comparable immunohistochemical studies on PGRMC2 have not been conducted.

Quantitative PCR measurements confirm that PGRMC1 expression increases as ovarian tumors progress from stage 1 in which the cancer cells are restricted to the ovary and fallopian tubes through stage 4 when the ovarian cancer cells have metastasized to various sites throughout the body (https://www.cancer.org/cancer/ovarian-cancer/detection-diagnosis-staging/staging.html, accessed on 15 September 2021). Specifically, the expression of PGRMC1 nearly doubles during this transition (Figure 1E) [4]. The subcellular localization of PGRMC1 also changes from mainly membrane and/or cytoplasmic to nuclear with increasing stage progression [4]. These changes in PGRMC1 expression levels and subcellular localization are consistent with the finding that patients with high tumor levels of PGRMC1 have a poorer prognosis and shorter disease-free state [41] (Figure 1F). It is important to appreciate that over-expression of other genes (e.g., HSPA1A, CD99, RAB3A, and POM121L9P) has also been associated with a shorter disease-free state and shorter overall survival [42]. These genes encode proteins that function as intracellular chaperones and could direct the subcellular distribution of PGRMC1 and PGRMC2 and likely contribute to ovarian cancer progression.

Not only are PGRMC1 and PGRMC2 expressed in ovarian cancer, but the nuclear progesterone receptor (PGR) is also expressed [4]. Given the expression of these receptors, it is not surprising that P4 plays a role in regulating the growth of ovarian cancers, with some studies indicating that P4 stimulates the growth of these cancers (for review see Ponikwicka-Tyszko et al. [5]). The concept that P4 promotes ovarian cancers resulted in several clinical trials in which the ability of the PGR antagonist, Mifepristone (a.k.a. RU486), to attenuate ovarian cancer progression was assessed. These trials basically failed to demonstrate that this PGR antagonist was effective in inhibiting ovarian cancer progression [45]. The reasons why Mifepristone failed to inhibit ovarian cancer are unknown and could be due to numerous factors. One important factor is that PGR is expressed at relatively low levels in both tumors [45] and cell lines derived from ovarian cancers (Expression Atlas, http://www.ebi.ac.uk/gxa, accessed on 15 September 2021). In addition, the expression of PGR decreases and PGRMC1 increases with ovarian cancer progression (Figure 1E) and as a result the ratio of PGRMC1 to PGR dramatically increases with the stage of ovarian cancer [46]. Finally, Mifepristone could actually promote ovarian tumor growth via a PGRMC1-dependent mechanism since the PGRMG1 antagonist, AG 205, has been shown to attenuate Mifepristone growth-promoting actions [5]. Given these findings, it is possible that the ability of P4 to promote the growth of ovarian tumors is mediated by PGRMC1 and possibly via an interaction with PGRMC2 as discussed in Section 6.1. However, it is important to appreciate that AG 205 has been shown to have PGRMC1-independent actions [47,48] so whether Mifepristone acts through PGRMC1 requires a more complete examination.

### 3.3. Cellular Localization and Expression Levels within Neoplastic Endometrium

The first published evidence of PGRMC1 expression in the endometrium was demonstrated in a microarray study and confirmed by PCR and Northern blot analyses in which PGRMC1 mRNA expression was found to be substantially higher in the estrogen-dominated proliferative phase compared to the P4-dominated secretory phase of the human menstrual cycle [49]. These studies were confirmed by a proteomic study in the human endometrium [50]. During the proliferative phase of the human menstrual cycle, PGRMC1 is abundantly expressed in both the epithelial and stromal compartments (Figure 2A), whereas PGRMC1 expression is greatly diminished in epithelial tissue of the secretory phase (Figure 2B). Given that PGRMC1 is most abundant during the proliferative phase of the menstrual cycle, a period dominated by ovarian-derived estradiol, PGRMC1 may play a role in regulating epithelial cell proliferation. Conversely, PGRMC2 expression remains relatively unchanged in epithelial cells from the proliferative to secretory phases of the human menstrual cycle (Figure 2D,E), but becomes elevated in stromal tissue during the secretory phase (Figure 2E). Endometrial studies also show that the expression of PGRMC1 (epithelium) and PGRMC2 (stroma) is tightly linked to the endocrine influences of estradiol and P4 during the menstrual cycle. Interestingly, the epithelial cells of the endometrial cancer samples express elevated levels of both PGRMC1 (Figure 2C) and PGRMC2 (Figure 2F) in a manner that more closely mimics that of samples obtained from the proliferative phase of the menstrual cycle.

Taken together, these expression profiles and cellular localization studies demonstrate that both PGRMC family members are expressed and likely influence the fate of ovarian and endometrial cancers. Therefore, it is important to discuss the various factors that regulate the expression and localization of PGRMC1 and PGRMC2.

## 4. Regulation of PGRMC1 and PGRMC2 Expression in Ovarian and Endometrial Cancers

### 4.1. Hormonal Regulation of PGRMC1 and PGRMC2

Not much is known about the hormonal regulation of PGRMC1 and PGRMC2 in neoplastic cells. However, in the ovary PGRMC1 expression changes during the estrous cycle. Prior to the luteinizing hormone (LH) surge, PGRMC1 is expressed in the granulosa cells of various sized follicles [51]. The LH surge induces ovulation and an increase in follicular PGRMC1 expression [51]. Moreover, LH receptors are highly expressed in the majority of ovarian cancers with expression levels being correlated with adverse outcomes [52]. Numerous signal transduction pathways are activated by LH that result in enhanced P4 synthesis as well as the sequential expression of numerous genes, including PGR [53]. One early study suggests that PGR suppresses PGRMC1 (a.k.a. 25-Dx) expression [54] as PGRMC1 levels in the ventral medial nuclei of PGR knockout mice are increased over controls. PGR could mediate its suppressive action through the glucocorticoid response elements (GRE) within the PGRMC1 promoter [55] given the structural similarity between GRE and the progesterone receptor response element [56]. If so, then the progressive decrease in PGR that is associated with the increasing stage of the ovarian cancer could account for the corresponding increase in PGRMC1 expression (Figure 1E,F) [4].

As previously indicated, uterine expression of PGRMC1 and PGRMC2 is also affected by steroid hormonal fluctuations that occur during the menstrual/estrous cycle. Previous studies reveal that P4 induced a 2-fold increase in both PGRMC1 and PGRMC2 mRNA in ovariectomized mice. Further, immunohistochemical studies showed that PGRMC1 is present in the stromal cells as well as the glandular and luminal epithelial cells with the PGRMC1 staining being predominant in the luminal epithelium of mice when serum P4 levels are elevated [57]. As shown in Figure 2, PGRMC1 and PGRMC2 change in the human endometrium in response to the menstrual cycle. Specifically, PGRMC2 becomes elevated in endometrial stromal tissue during the P4-dominated secretory phase of the cycle, while PGRMC1 expression is most evident in the luminal and glandular epithelia of the proliferative phase and down regulated in these tissues during the secretory phase. These observations are consistent with the concept that P4 is involved in regulating the expression of both PGRMC1 and PGRMC2 in the human endometrium with the change in P4-directed expression mediated in part by PGR and PGRMC1 [58]. Since PGR is only expressed in about 50% of endometrial cancers [59], the loss of PGR could lead to an increase in PGRMC1 as suggested for ovarian cancer. This must be demonstrated by further experimentation but the findings that 1) there is an inverse relationship between the expression of PGRMC1 and PGR and 2) PGRMC1 levels increase while PGR levels decrease with the increasing grade of endometrial cancers support this concept [60].

Hormones can regulate PGRMC1 expression by not only regulating the expression of transcription factors but also microRNAs (miRNAs). miRNAs are small non-coding RNAs that bind to mRNA and most often target their destruction, thereby playing an important role in regulating gene expression in various cancers [61]. An analysis of the 3′-UTR of PGRMC1 using TargetScan detected two putative Let-7i/miR-98 sites and one miR-200a/141 binding site [62]. Treatment with the Let-7 mimic suppressed PGRMC1 mRNA levels in the ovarian cancer cell line, SKOV-3. The functionality of the Let-7i site was assessed using a luciferase reporter construct whose expression was directed by the 3′-UTR region of PGRMC1. These studies revealed that over-expression of Let-7i suppressed luciferase activity that was dependent on the Let-7 site. However, neither miR-141 nor miR-200a affected the levels of luciferase activity. Interestingly, in the ovarian cancer cell line, SKOV-3, P4 increased the levels of Let-7 and decreased the levels of PGRMC1 mRNA [62]. As Let-7i is down regulated in ovarian cancer [63], the decrease in Let-7i expression in ovarian cancer could be part of the mechanism that accounts for an increase in PGRMC1 expression. Consistent with this concept, Liu et al. [64] demonstrated that Let-7i delivery to ovarian cancer cells (i.e., OVCAR3 cells) reversed the paclitaxel resistance in part by decreasing PGRMC1 levels.

Similar studies were conducted using Ishikawa cells, a cell line derived from an endometrial adenocarcinoma, and revealed that transfected miR-98, a Let-7 family member [65], suppressed PGRMC1 expression through a direct interaction with the 3′-UTR region of PGRMC1 [60]. Taken together, these studies demonstrate that the Let-7 family of miRNAs can directly influence the abundance of PGRMC1 mRNA.

### 4.2. Environmental Regulators of PGRMC1 and PGRMC2

In addition to hormones, transcription factors, and miRNAs, environmental factors can also influence PGRMC1 expression, one of which is dioxin [66,67]. Polychlorinated dibenzo-p-dioxins (i.e., dioxins) are one of many polycyclic hydrocarbons that were used as a tactical herbicide (e.g., Agent Orange) and are persistent organic pollutants. While 2,3,7,8-tetrachlorodibenzo-p-dioxin (TCDD) is the most toxic dioxin in mammalian cells, it is not itself mutagenic or genotoxic. Rather, it promotes tumor progression induced by other compounds. The endocrine-disrupting actions of dioxins, TCDD in particular, are commonly mediated by the aryl hydrocarbon receptor (AHR). The promoter region of PGRMC1 has several AHR/ARNT sites, which could be bound by dioxin-activated AHR to elevate the expression of PGRMC1. In this light, dioxin-induced expression of PGRMC1 could account for much of dioxin’s ability to promote tumor development, since PGRMC1 functions to promote cell survival [4] as will be discussed later in this review. In addition to dioxins, exposure to environmental pollutants can lead to an increase in PPARγ [68]. PPARγ is elevated in both ovarian [69] and endometrial [70] cancers and ligand activation of PPARγ increases PGRMC1 in adipocytes [71], making it likely that PPARγ is also involved in increasing PGRMC1 levels in reproductive cancer.

## 5. PGRMC1 and PGRMC2 as Regulators of Tumor Growth

The previous studies have demonstrated that PGRMC1 is highly expressed in ovarian and endometrial cancers and that its expression is regulated by both hormonal and environmental factors. However, these studies are limited in that they do not provide insight into PGRMC1′s functional role in cancer, specifically as it relates to tumor growth and chemoresistance.

### 5.1. In Vivo Studies Using Human Ovarian and Endometrial Cancer Xenografts

In order to better define the factors that regulate tumor growth and to develop therapeutics to attenuate cancer development and metastasis, various in vivo models have been developed as reviewed by Magnotti and Marasco [72]. The most common models are xenografts generated by injecting human cancer cells into immunodeficient mice. More recently, orthotopic models have been developed in which tumor cells are transplanted into their organ of origin. These models place tumor cells in an environment that more closely mimics the disease state. In the case of ovarian cancer, the cancerous cells/tissue is placed under the bursa that surrounds the ovary. Along with the orthotopic models, xenograft models using patient-derived tumors, referred to as patient-derived xenograft (PDX) models, have been developed. The PDX models offer the possibility of assessing individual patient cancer cell growth, tumor progression, and sensitivity to various chemotherapies. Finally, there are humanized mouse models that express human immune cells. Like the PDX models, the humanized mouse models have enhanced potential to reveal novel ovarian cancer immunotherapies.

Unfortunately, the functional role of PGRMC1 has not been assessed using these more sophisticated mouse models. However, as outlined below, xenograft tumors have been used to reveal PGRMC1′s role in regulating growth parameters, vascular genesis, and response to chemotherapeutics in ovarian and endometrial tumors.

#### 5.1.1. Ovarian Cancer Xenografts

To date, there has been only one xenograft model developed to assess PGRMC1′s role in ovarian cancer [44]. This model used SKOV-3 cells, a human ovarian cancer cell line.

SKOV-3 cells do not express PGR but express high levels of PGRMC1, an expression profile that mimics human ovarian cancer as previously described. SKOV-3 cells were modified to constitutively express the fluorescent marker, dsRed, to allow for the localization of SKOV-3 cell tumors to be assessed in vivo. Importantly, expressing dsRed neither affected the sensitivity of the SKOV-3 cells to undergo apoptosis in response to cisplatin (CDDP) nor the ability of P4 to inhibit CDDP-induced apoptosis in vitro as will be discussed further in Section 6.2.1. The dsRed-SKOV-3 cells were then depleted of PGRMC1 using an shRNA approach and injected into the peritonea of athymic nude female mice. Tumors derived from parental PGRMC1-intact and PGRMC1-depleted SKOV-3 cells developed within 5 weeks. However, fewer mice injected with PGRMC1-depleted SKOV-3 cells developed tumors. Those that did had fewer tumors that were smaller (Figure 3A inset) than those derived from PGRMC1-intact SKOV-3 (Figure 3A). PGRMC1-depleted ovarian tumors also had fewer apoptotic cells as assessed by TUNEL assay and unlike the PGRMC1-intact controls failed to show an increase in the CDDP-induced apoptosis (Figure 3B). They also had poorly developed microvasculature (compare Figure 3C with 3D) [73] and higher levels of HIF1 and ARNT (Figure 3E), suggesting they were in a hypoxic environment, which is likely associated with a metabolic shift to glycolysis. This would be consistent with the finding of Sabbir that P4-PGRMC1 signaling alters cellular respiration known as the Warburg effect [74].

#### 5.1.2. Endometrial Cancer Xenografts

Aside from one study suggesting that PGRMC1 expression increases with endometrial tumor grade and that PGRMC1 may be directly repressed by miRNA-98 in cancer cell lines [60], very little is known about a role for PGRMC proteins in the development and progression of endometrial cancer. To address this issue, endometrial xenograft tumors derived from Ishikawa cells were developed in intact female NOD/SCID mice. PGRMC1-intact control Ishikawa cells and shRNA infected PGRMC1-deplete cells were injected subcutaneously into these mice [3].

This study revealed that the endometrial xenograft tumors derived from PGRMC1-intact cells grew much faster than tumors derived from PGRMC1-deplete cells (Figure 4A) and were significantly heavier (Figure 4B). Similar findings were observed when PGRMC1-intact and PGRMC1-deplete intraperitoneal tumors were examined (Compare Figure 4C with 4D). Little difference in tumor volume was observed in PGRMC1-intact cells treated with chemotherapy (i.e., paclitaxel followed by CARBOplatin). In contrast, chemotherapy-treated PGRMC1-deplete tumors had a significantly smaller volume compared with vehicle-treated tumors (Compare Figure 4E with 4F). The decrease in tumor volume could not be explained by poor nutrition or weight loss since no difference in mouse weight occurred in any of the mice throughout the 15-day time period [3]. A similar observation was made in PGRMC1-intact versus PGRMC1-deplete cells treated with doxorubicin in vitro, in which PGRMC1-deplete cells were significantly more sensitive to chemotherapy treatment than PGRMC1-intact cells [3]. Collectively, these studies illustrate the growth-promoting and cell stress protective functions of PGRMC1 in endometrial cancer cells.

A role for PGRMC2 in the development and progression of endometrial cancer was also assessed using a genetic approach in vivo. In this issue of *Cancers*, Kelp et al. used a conditional *Pten* loss-of-function mouse model of endometrial cancer to demonstrate that *Pgrmc2* deficiency attenuated the development of endometrial hyperplasia and cancer and prolonged survival.

Taken together, these xenograft studies clearly demonstrate that PGRMC1 plays important roles in growth and chemosensitivity of both ovarian and endometrial tumors. Why ablating PGRMC1 renders ovarian cancer resistant to chemotherapy while enhancing the sensitivity of endometrial cancers remains to be determined. In addition, more precise insights into the role that these PGRMCs play in chemosensitivity could be provided by studies utilizing xenografts derived from specific patients (i.e., PDX) [75], since some of these PDX models likely express PGRMCs at various levels based on an analysis of PGRMC expression in various cancer subtypes [41]. Moreover, these PDX models have the potential to evaluate the effectiveness of agents that might interfere with PGRMCs’ survival function as suggested by Ponikwicka-Tyzko et al. [5]. However, such in vivo models have a limited ability to identify specific biological and cellular functions that are regulated by PGRMC1. Identifying these actions is better suited for in vitro approaches.

## 6. Biological Functions of PGRMC1 and PGRMC2

PGRMC1 and PGRMC2 are involved in regulating numerous biological processes including mitosis, apoptosis, cell migration, and chemoresistance. Although details regarding the mechanisms by which PGRMC1 and PGRMC2 coordinate each of these biological functions in either normal or neoplastic tissues are limited, it appears that the effects of PGRMC1 and PGRMC2 are mediated by both membrane-initiated events and nuclear modes of action.

### 6.1. PGRMC1 as a Regulator of Mitosis and Apoptosis

PGRMC1 and PGRMC2 appear to control the rate of mitosis by regulating two aspects of cell division: stability of the mitotic spindle and entry into the cell cycle. Previous studies have demonstrated that in an ovarian cancer cell line, SKOV-3 cells, PGRMC1, and PGRMC2 interact with tubulin to regulate the stability of the mitotic spindle [76]. This will be discussed in detail in the review by Dr. Valentina Lodde and associates to be published in this issue of *Cancers*. As indicated, PGRMC family members also regulate entry into the cell cycle as initially demonstrated in spontaneously immortalized granulosa cells. In these studies, depleting either PGRMC1 or PGRMC2 resulted in a rapid increase in the rate at which cells enter the cell cycle as judged by BrdU incorporation or the expression of the G_1_/S component of the FUCCI cell cycle sensor [77]. Interestingly, entry into the cell cycle that was initiated by depleting either of these PGRMC family members did not lead to an increase in cell number but rather apoptotic cell death. In addition, the absence of PGRMC1 or PGRMC2 eliminated P4′s ability to suppress entry into the cell cycle [77]. These findings were subsequently confirmed in cultured human granulosa/luteal cells [37].

The finding that depletion of either of these PGRMC family members initiated entry into the cell cycle but terminated in apoptotic cell death raises the possibility that their expression changes depending on the stage of the cell cycle. This appears to be the case as a decrease in PGRMC2 levels but not PGRMC1 is observed in cells in the G_1_/S phase of the cell cycle. This decrease in PGRMC2 is transient, returning to G_0_ stage levels by the G_2_/M stage of the cell cycle [78]. This transient change in PGRMC2 likely explains why the entry into the cell cycle is normally followed by cytokinesis and not apoptosis.

The mechanism through which depleting either PGRMC1 or PGRMC2 leads to entry into the cell cycle is related to a complex interaction between GTPase-Activating Protein Binding Protein 2 (G3BP2) and either PGRMC1 or PGRMC2. These interactions were demonstrated by pulldown assays, colocalization, and in situ proximity assays [77]. That G3BP2 is an essential component of the PGRMC1 and PGRMC2 complexes, which limits entry into the cell cycle, was confirmed by the finding that depleting G3BP2 leads to a premature entry into the cell cycle even in the presence of PGRMC1 and PGRMC2. Interestingly, the only known protein that interacts with G3BP2 is the inhibitor of nuclear factor of kappa light polypeptide gene enhancer in B-cells inhibitoralpha (IκBα), which binds to the transcription factor, nuclear factor of kappa light polypeptide gene enhancer in B-cells (NFκB; RelA), restricting its localization to the cytoplasm. Depleting G3BP2 induces the translocation of NFκB to the nucleus, initiates NFκB-dependent transcriptional activity, and triggers entry into the cell cycle [79]. How P4 affects this pathway is unknown but the possibility exists that P4 increases the affinity of the PGRMC1:PGRMC2:G3BP2:IκBα complex for NFκB, thereby maintaining NFκB’s cytoplasmic localization. This putative increase in affinity in response to P4 could be due to changes induced by P4 binding directly to PGRMC1 [78] or by P4 binding to PAQR7 (a.k.a. membrane progestin receptor alpha), which in turn binds to PGRMC1 [80].

Whether this PGRMC1/PGRMC2-dependent pathway functions to regulate the proliferation of ovarian and endometrial cancer cells remains to be determined. This putative mechanism may be conditioned on the cell type and/or various endocrine factors since depletion of PGRMC1 in lung cancer cell lines results in a decrease in NFκB activity [81].

This PGRMC1:PGRMC2-based mechanism is one of the best described membrane-initiated mechanisms of action, which regulate mitosis and apoptosis. There are likely other membrane-initiated mechanisms that involve various kinases such as PI3/AKT signaling [81], activation of the Wnt/ß-catenin pathway [82], and the EGF receptor [83]. These are discussed in more detail in reviews by Dr. Michael Cahill [10,38].

Membrane-initiated actions appear to be mediated by the monomeric form of PGRMC1 as it localizes to the cytoplasm [36]. However, oligomers of PGRMC1 localize to the nucleus suggesting that PGRMC1 has a nuclear site of action [36] as will be discussed in Section 7.

### 6.2. PGRMC1 and Chemoresistance

There are effective surgical and chemotherapeutic treatments for both ovarian and endometrial cancers, especially if these cancers are treated in their early stages with the most common chemotherapies including cisplatin or its derivatives [84]. However, over time these cancers frequently reoccur. It appears that chemotherapy can induce epithelial to mesenchymal transition that is accompanied by changes in several signal transduction pathways, reduced expression of the adhesion protein, E-cadherin, and subsequent metastasis. These events are often associated with chemoresistance [84]. The mechanism responsible for chemoresistance remains to be clearly defined, likely because multiple mechanisms regulate the responsiveness to chemotherapeutic drugs including the DNA repair mechanism [75]. Two mechanisms of importance are the presence of cancer stem cells and the ability to metastasize [84,85]. The studies presented in the following paragraphs outline a potential role for PGRMC1 and PGRMC2 in regulating both metastasis and cancer stem cell function.

#### 6.2.1. PGRMCs and Preserving Responsiveness to Cisplatin

As indicated, ovarian [86] and endometrial cancers [87] develop in a hypoxic environment in which vascular development is limited and the expression of genes that control numerous signal transduction and survival pathways are altered [88]. One example of changes that occur in both ovarian and endometrial cancers is an upregulation of efflux pumps that remove chemotherapeutic drugs from the cancer cells [84]. Little is known as to how these changes are initiated. Insight into the involvement of PGRMC1 in regulating efflux pumps/transporters was obtained from xenograft studies in which ovarian tumors were derived from SKOV-3 cells in which PGRMC1 was depleted (Figure 3E) [44]. Specifically, mRNA levels of several ATP-binding cassette (ABC) transporters were increased several-fold in ovarian tumors derived from PGRMC1-deplete SKOV-3 cells compared to parental SKOV-3 cells. Importantly, the ABCC2 transporter was expressed at > 3-fold higher in PGRMC1-deplete tumors compared to control tumors and this transporter is highly effective in removing cisplatin from human cancer cells [73]. Thus, even though they grow more slowly, PGRMC1-deplete ovarian tumors express several ABC transporters, which likely accounts for their greater resistance to cisplatin.

Similar studies were conducted using xenografts derived using PGRMC1-intact and PGRMC1-deplete endometrial tumors [3]. These studies demonstrated that the PGRMC1-deplete tumors grew more slowly than those generated using PGRMC1-intact endometrial cancer cells. Moreover, treatment with paclitaxel followed by CARBOplatin decreased the weight of the PGRMC1-deplete tumors by 4-fold compared to the tumors derived from PGRMC1-intact cells. Taken together, these xenograft studies demonstrate that the presence of PGRMC1 differentially affects the ability of various chemotherapeutic agents to destroy ovarian and endometrial tumors. This clearly reflects the complexity involved in assessing the relationship between expression of PGRMC1 and cancer-specific responses to various chemotherapeutic agents.

#### 6.2.2. PGRMCs and Cancer Stem Cells

Why some cancer cells are resistant to various chemotherapeutic agents has not been clearly resolved. Although this remains a controversial area of research, there is evidence that both ovarian [89] and endometrial [90] tumors are composed of chemosensitive cells and cancer stem cells that maintain tumor chemoresistance. One factor is that these cancer stem cells typically reside in a quiescent state, which is important in that chemotherapeutic drugs are generally intercalating or alkylating chemicals, antimetabolites, or mitotic inhibitors that more effectively target rapidly dividing cells (see review by Li et al. [89]). As a result, the cancer stem cells remain viable after chemotherapy and can undergo asymmetric cell division thereby generating both a new stem cell and a more differentiated tumor cell.

This process of self-renewal can be promoted by stem cell growth factor receptor, c-Kit, which is a tyrosine kinase oncoprotein that is expressed by ovarian [89] and endometrial [90] cancer stem cells. Moreover, c-Kit expression is induced by HIF1A [89], known to be highly expressed in cancers as they are in a hypoxic environment [86,88]. Although endometrial cancer stem cells have not been examined, ovarian cancer stem cells express PGRMC1 [91], which also stimulates stem cell self-renewal [82]. PGRMC1 may promote stem cell self-renewal by stimulating the expression of Kit ligand. This is based on the finding that depleting PGRMC1 and PGRMC2 decreases the expression of Kit ligand inmouse ovaries [92]. Whether a similar relationship between PGRMC1/PGRMC2 and Kit ligand expression exists in ovarian and/or endometrial cancer stem cells remains to be determined.

#### 6.2.3. PGRMCs and their Role in Cell Migration and Metastasis

Even after surgical debulking and/or chemotherapy, ovarian cancer recurs in about 70% of patients often due to metastasis [93]. Cell migration and the resulting formation of metastatic lesions limits the ability to remove the cancers surgically, thereby restricting treatment to chemotherapy and/or radiation therapy. Unfortunately, chemotherapy with platinum- and paclitaxel-based drugs may induce a cascade of ill-defined changes that actually facilitate cell migration [93]. In vitro ovarian cancer cell migration is stimulated by miR-21, which is highly expressed in ovarian tumors [94]. Similarly depleting PGRMC2 but not PGRMC1 also increases ovarian cancer cell migration [95]. How these in vitro studies relate to the capacity of ovarian cancer cells to metastasize after chemotherapy is unknown. Since miR-21 levels are increased in cisplatin-resistant ovarian cancer cell lines compared to cisplatin-responsive ovarian cancer cells [93] and cisplatin treatment in vitro reduced PGRMC2 levels [95], it is possible that reduced levels of PGRMC2 combined with miR-21 regulation of signaling cascades, which are activated after chemotherapy, facilitates the metastasis of ovarian cancer. These in vitro studies suggest that therapeutic manipulation of the ratio of miR-21 to PGRMC2 could provide a means to limit the metastasis, but this concept must be further tested using in vivo models.

Endometrial cancers rarely metastasize [96]. The reason for this is unclear but one study offered the suggestion that PGRMC2 may function to suppress tumor metastasis of endocervical adenocarcinoma. This was based on a comparative genomic hybridization array analysis, where loss of PGRMC2 was prominently associated with cases of nodal metastasis [97]. As with ovarian cancers, miR-21 levels in endometrial cancers are on average 2-fold higher than in benign tissue [98], which is consistent with the concept that elevated miR-21 in the presence of low levels of PGRMC2 promotes migration of both ovarian and endometrial cancers. However, PGRMC2′s role in suppressing metastasis may be cancer-specific since Ye et al. [96] demonstrated in breast cancer cells that knockdown of PGRMC2 actually inhibited cell migration.

## 7. PGRMCs and the Regulation of RNA and Protein Synthesis

### 7.1. PGRMC1 and its Nuclear Site of Action

In addition to membrane-initiated events as outlined in Section 6 of this review, PGRMC1 also localizes to the nucleus, where it is likely involved in regulating RNA synthesis and in turn cell proliferation and/or cell survival as well as chemoresistance. This is supported by the observation that PGRMC1 localizes to the nucleus in more advanced and rapidly growing ovarian tumors [4]. Interestingly, in spontaneously immortalized granulosa cells, PGRMC1 is detected as a monomer (i.e., ≈ 25 kDa) in the cytoplasm while oligomers are localized to the nucleus [36]. The capacity to form dimers and oligomers is an important factor in PGRMC1′s action. PGRMC1 dimers form via an interaction between heme molecules that project from adjacent PGRMC1 molecules [99]. The heme:heme interaction is essential for various aspects of PGRMC1′s actions, including the interactions with EGF receptors and cytochromes P450 [99]. Moreover, P4 binds to PGRMC1 at the putative heme/ligand binding cleft and induces conformational change to the heme [99]. How P4 binding influences the configuration of PGRMC1 and therefore its biological function remains to be fully elucidated.

Further support for a genomic mode of action for PGRMC1 comes from studies in which transfected PGRMC1-GFP localizes to the cytoplasm and increases the sensitivity to P4′s anti-apoptotic actions, but only in the presence of endogenous PGRMC1, which is present in both the cytoplasm and nucleus. However, if endogenous cytoplasmic and nuclear PGRMC1 are depleted, then PGRMC1-GFP no longer enhances P4′s actions [36]. Additional support for a nuclear site of action for PGRMC1 is provided by the finding that P4′s anti-apoptotic action is dependent on de novo RNA synthesis [100].

How PGRMC1 functions within the nucleus is not clearly defined. It is known that nuclear PGRMC1 is sumoylated and that P4 increases the sumoylation of PGRMC1 [36]. That sumoylation is an important component of PGRMC1′s action is revealed by the observations that mutating PGRMC1′s sumoylation sites increases TCF/LEF transcriptional activity and attenuates P4′s ability to suppress TCF/LEF transcriptional activity [36]. How nuclear PGRMC1 affects transcription is not known but ChIP studies failed to reliably detect an interaction with DNA. This suggests that the more likely mode of action would be that PGRMC1 sequesters various transcription factors thereby modulating their ability to regulate transcription. This hypothesis, however, requires considerably more investigation. Regardless of the mechanism of action, various transcriptome analyses have demonstrated that the presence of PGRMC1 affects the gene profile in a cell-type-dependent manner [101].

### 7.2. PGRMCs and Ribosomal Protein Interaction

In addition to altering the mRNA expression profile, PGRMC1 and PGRMC2 are likely involved in determining which mRNAs get translated. This concept was initially proposed by Cahill et al. [38] based on the observation that PGRMC1 localizes to the nucleolus, the site of ribosome biogenesis. Cahill et al. [38] further speculated that PGRMC1 could be interacting with “ribosomes,…”. Prior to Cahill’s speculation, we conducted a series of studies in which spontaneously immortalized granulosa (ovarian) cells were transfected with expression vectors that encode either GFP-tagged PGRMC1 or GFP-tagged PGRMC2 fusion proteins that were subsequently used in “pull down” experiments. Proteomics was employed to identify proteins that interacted with each PGRMC family member. The DAVID tool was used to identify proteins that were significantly enriched for specific biological categories. This analysis revealed that proteins associated with PGRMC1 and PGRMC2 were most frequently observed in the ribosomal/protein synthesis category (Enrichment score 8.36). The identified proteins included but were not limited to small (40s) and large (60s) ribosomal proteins; the most abundant of these are listed in Table 1. While most of the ribosomal proteins interacted with both PGRMC family members, a few ribosomal proteins preferentially interacted with only one of the PGRMC family members (Table 2). In a follow-up study, spontaneously immortalized granulosa cells were transfected with expression vectors that were Flag-tagged PGRMC1 (i.e., Wild type) or Flag-tagged PGRMC1 in which serine 57 and 181 were changed to alanine (i.e., ∆PO_4_).

This study revealed that the ability of some ribosomal proteins was dependent on the phosphorylation status of serine 57 and/or 181 of PGRMC1 interactions with ribosomal proteins were also observed in pulldown experiments involving endometrial cancer cells with PGRMC1 interacting with many ribosomal proteins in endometrial cancer cell lysates (Table 3). It is anticipated that the proteomic data will be published as a separate paper with complete details.

Ribosomes are highly conserved and composed of 40s and 60s complexes that serve as the site of protein synthesis [102,103]. Briefly, transfer RNA interacts within a 40s ribosomal complex, which includes the initiation factors EIF2 and EIF3. This complex attracts the capped 5′-end of mRNA. Once the start codon is detected, a 60s ribosomal component enters the ribosomal complex and the translation of mRNA into protein is initiated [102,103]. Although the ribosome is always composed of a 40s and a 60s component, the ribosomal proteins that make up each of the ribosomal components vary greatly from one cell type to another, as well as in normal versus disease states [104]. This variation has led to the concept that the precise ribosome composition acts as a “filter” and “selects” specific mRNAs to be incorporated into the ribosome [105]. This filtering mechanism determines in part which mRNAs are translated and which are not. It has been proposed that the selective nature of the “ribosomal filter” accounts for the discord in the relative level of mRNA and protein that is often observed (correlation coefficient ≤ 0.50) [106,107].

In mammals, 36 and 57 proteins or proteins with posttranslational modifications have been identified as members of the 40s and 60s ribosomal protein families, respectively [104]. Interestingly, in ovarian cells PGRMC1 and PGRMC2 each bind 8 of these ribosomal proteins, while PGRMC2 binds 2 additional ribosomal proteins and PGRMC1 binds 2 additional ribosomal proteins found in ovarian isolates. Since PGRMC1 and/or PGRMC2 interact with 10 of the 93 ribosomal proteins or post-translationally modified ribosomal proteins (≈ 10% of ribosomal proteins), these PGRMC family members may be present in a limited, select set of ribosomal complexes. Based on the “ribosomal filter” concept, we propose that PGRMCs regulate the translation of a very specific subset of proteins whose synthesis is directed by ribosomes composed of ribosomal proteins that interact with PGRMC1 and/or PGRMC2. However, this concept remains to be rigorously tested.

The PGRMCs could regulate ribosomal composition in two ways. One way would be to act as scaffolding proteins that would bring specific ribosomal proteins and other translation-related proteins together to promote synthesis of specific proteins. A second way could be that PGRMCs bind specific ribosomal proteins and restrict their access to ribosomal complexes. In addition, P4 may modulate the PGRMC1/PGRMC2 interactions with ribosomal proteins. Whether the abilities of PGRMC1 and PGRMC2 to influence the composition of the ribosomes and thereby which RNAs get translated into specific proteins changes at different stages of ovarian cancer remains to be determined.

## 8. Conclusions and Future Directions

This review has summarized the important roles that PGRMC1 and PGRM2 play in regulating the developmental fate of both ovarian and endometrial tumors by presenting data on the expression of these PGRMC family members and the functional relationship between PGRMC expression and various biological actions as revealed using various genetic approaches. However, there are still voids in our understanding of their mechanism of action. Specifically, it will be essential that studies be conducted to determine how (1) PGRMCs function within each of their subcellular sites, (2) various posttranslational modifications influence PGRMCs’ biological action, (3) PGRMCs regulate cancer stem cell function and chemoresistance and (4) PGRMCs interact with ribosomes to regulate which mRNAs get translated into proteins. Addressing these issues could provide sufficient detail so PGRMC agonists and antagonists could be developed into adjunct chemotherapeutic agents to treat ovarian and endometrial cancer and potentially many other cancers that express PGRMC family members.

## Figures and Tables

**Figure 1 cancers-13-05953-f001:**
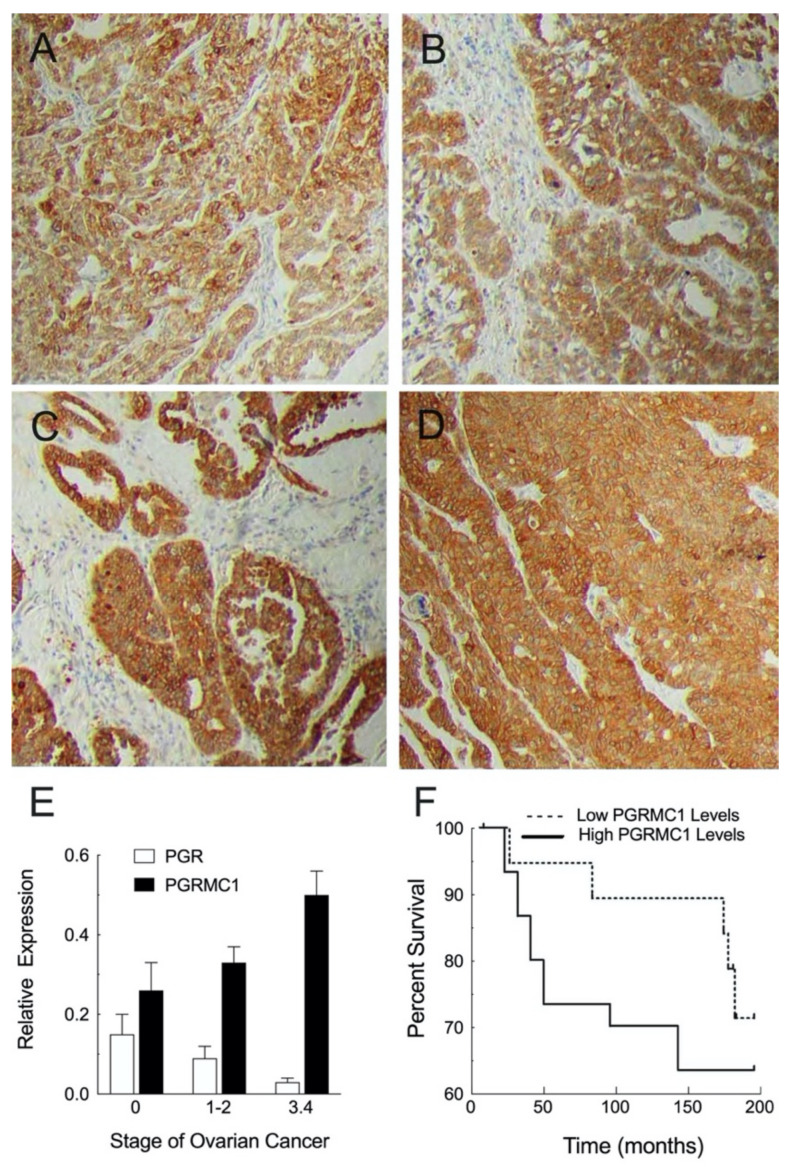
The localization of PGRMC1 in ovarian cancers: clear cell carcinoma (**A**), endometroid carcinoma (**B**), papillary serous cystadenocarcinoma—stage IIB (**C**), and papillary serous cystadenocarcinoma stage IIIC (**D**). The relationship between the stage of ovarian cancer and the expression of PGR and PGRMC1 is shown in panel **E**, while the relationship between the level of PGRMC1 and patient survival is shown in panel **F**. Note that defining the stages of ovarian cancer is evolving based on molecular markers [43] but the staging used to identify the stages of the images and graph in this Figure is based on the FIGO stages as described on the American Cancer Society website (https://www.cancer.org/cancer/ovarian-cancer/detection-diagnosis-staging/staging.html, accessed on 15 September 2021). Images in panels **A**–**D** were not modified and are taken from Supplemental Figure 2 of the paper published by Hampton et al. [41] (https://www.ncbi.nlm.nih.gov/pmc/articles/PMC5113835/, accessed on 15 September 2021) and the graph in panel F was not modified and was taken from Figure 2a in the article by Hampton et al. [41] licensed under CC BY-4.0 (https://creativecommons.org/licenses/by/4.0/, accessed on 15 September 2021). Panel E is redrawn from Figure 1A,B from the article by Peluso et al. [44] with permission from CCC RightsLink.

**Figure 2 cancers-13-05953-f002:**
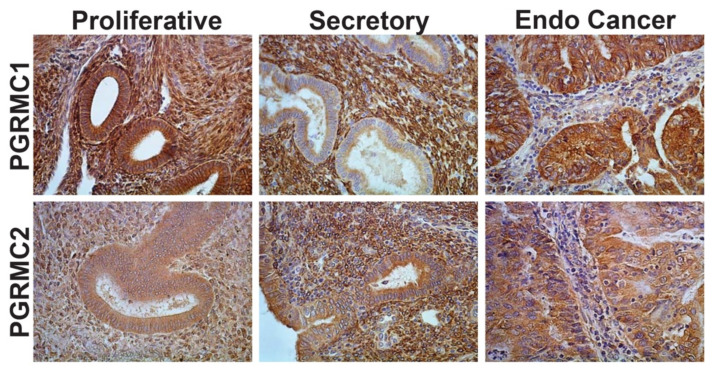
The localization of PGRMC1 and PGRMC2 in human endometrium collected during the proliferative and secretory phases of the menstrual cycle. PGRMC1 and PGRMC2 staining associated with endometrial (Endo) cancer is also shown. These proteins were detected by immunohistochemistry and are revealed by the brown stain (Pru, unpublished).

**Figure 3 cancers-13-05953-f003:**
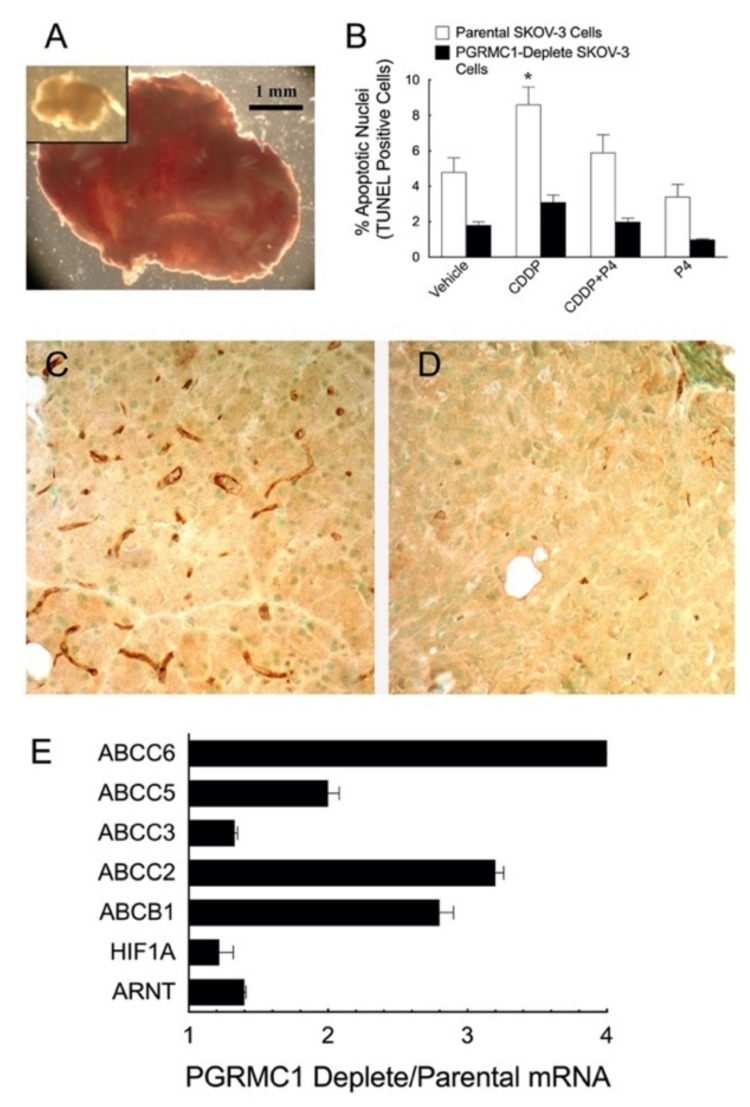
Comparison of xenograft tumor growth in PGRMC1-intact and PGRMC1-deplete dsRed-SKOV-3 cells. Panel **A** shows a typical tumor derived from PGRMC1-intact dsRed-SKOV-3 cells, while a typical tumor derived from PGRMC1-deplete dsRed-SKOV-3 cells is shown in the inset of panel **A**. Images are shown at the same magnification (Peluso, unpublished data). The effect of cisplatin (CDDP 8 mg/kg body weight, i.p.) and P4 (1 mg/0.1 mL of oil, s.c.) on the percentage of apoptotic nuclei is shown in **B**. * indicates a significant difference from control (*n* = 24 to 65 tumors/treatment; *p* < 0.05). The microvasculature of tumors derived from PGRMC1-intact (**C**) and PGRMC1-deplete (**D**) dsRed-SKOV-3 cells is revealed by staining for the endothelial cell marker, CD31 (brown stain). (**E**) Expression of ARNT, HIF1A, and several ABC transporters are shown as a ratio of the mRNA levels in PGRMC1-deplete to PGRMC1-intact ovarian tumors. All changes in mRNAs are statistically significantly different (*n* = 4; *p* < 0.05). Images and graphs taken from Figure 6 of the article by Peluso et al. [4] with permission from CCC RightsLink.

**Figure 4 cancers-13-05953-f004:**
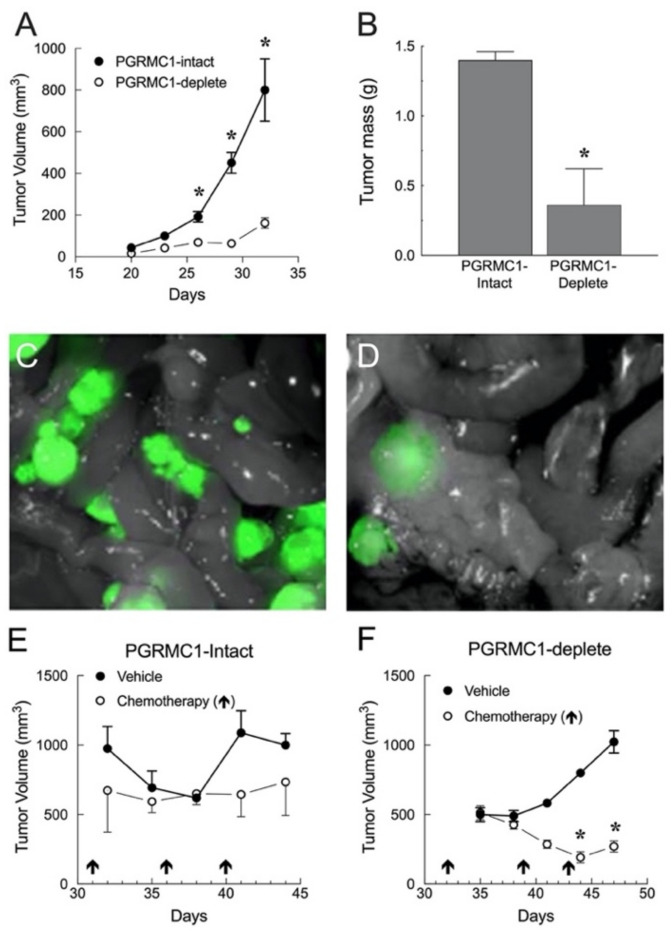
Comparison of xenograft tumor growth in PGRMC1-intact and PGRMC1-deplete endometrial tumor cells after a subcutaneous flank injection (**A**) or 6 weeks after an intraperitoneal injection (**B**). Images of intraperitoneal xenografts after injection with GFP-PGRMC1-intact (**C**) or GFP-PGRMC1-deplete (**D**) EV3 Ishikawa cells. Response of PGRMC1-intact (**E**) or PGRMC1-deplete (**F**) Ishikawa cell tumors treated with either vehicle (cremophor EL: ethanol) or paclitaxel and CARBOplatin injected as indicated by the arrows in plates (**E**) and (**F**). ***** indicates that a value is different from controls (*p* < 0.05). Data redrawn from Figure 6A–D and Figure 7A,B from an article by Friel et al. [3] with permission from CCC RightsLink.

**Table 1 cancers-13-05953-t001:** Identification of 40s small and 60s large ribosomal proteins that interact with GFP-PGRMC1 and GFP-PGRMC2 fusion proteins in spontaneously immortalized granulosa cells as per previously described protocol [77]. The list of proteins with each protein having at least 10% protein coverage and two or more unique peptides assigned to each protein. ND = not detectable (≤1 peptide fragment detected). (Peluso unpublished data.)

	PGRMC1	PGRMC2
Protein Name	% Coverage	% Coverage
ribosomal protein S4	14	10
ribosomal protein S26	31	ND
ribosomal protein S27	24	38
ribosomal protein L3	25	ND
ribosomal protein L4	17	28
ribosomal protein L12	18	32
ribosomal protein L27	ND	40
ribosomal protein L30	30	70
ribosomal protein L34	24	21
ribosomal protein L38	ND	50

**Table 2 cancers-13-05953-t002:** Identification of 40s small and 60s large ribosomal proteins that interact with Flag-PGRMC1 and Flag-PGRMC1 in which serine 57 and 181 were changed to alanine (i.e., ∆PO_4_) in spontaneously immortalized granulosa cells as per previously described protocol [36]. The list of proteins with each protein having at least 10% protein coverage and two or more unique peptides assigned to each protein. ND = not detected (≤1 peptide fragment detected). (Peluso unpublished data.)

	PGRMC1	PGRMC1-∆PO_4_
Protein Name	% Coverage	% Coverage
ribosomal protein S4	54	47
ribosomal protein S26	ND	44
ribosomal protein S27	ND	39
ribosomal protein L3	ND	45
ribosomal protein L4	46	51
ribosomal protein L12	ND	53
ribosomal protein L30	70	70
ribosomal protein L34	38	ND

**Table 3 cancers-13-05953-t003:** Identification of 40s small ribosomal proteins (RPS) and 60s large ribosomal proteins (RPL) that interact with PGRMC1 in Ishikawa endometrial cancer cells. Flag antibody coated magnetic beads were used to pull-down PGRMC1-FLAG:protein complexes in PGRMC1-FLAG expressing cells. The list of proteins is based on matched peptides identified using Proteome Discoverer software with each protein having at least 10% protein coverage and two or more unique peptides assigned to each protein.

Ribosomal Protein Interaction with PGRMC1 (% coverage)
RPS25 (30%)	RPL22 (30%)	RPS12 (54%)	RPL4 (24%)
RPS4X (40%)	RPS10 (15%)	RPS18 (20%)	RPS20 (19%)
RPS14 (32%)	RPL7A (28%)	RPS19 (51%)	RPSA (23%)
RPS17 (56%)	RPL5 (22%)	RPS23A (41%)	RPS3 (19%)
RPS15A (18%)	RPL18 (49%)	RPS13 (35%)	RPS11 (42%)
RPL13A (11%)	RPL38 (47%)

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
