# Peer review of "Progesterone Receptor Membrane Component (PGRMC)1 and PGRMC2 and Their Roles in Ovarian and Endometrial Cancer"

_cancers, 2021, doi:10.3390/cancers13235953_

Round 1

Reviewer 1 Report

In this review manuscript, the authors describe about comprehensive overview of the association between PGRMC1 and PGRMC2 in breast cancer. PGRMC1 is highly expressed in early stage of several types of breast cancer progression, and high expression of PGRMC1 in breast cancer potently decreases survival of patients. As a mechanism of cancer exacerbation mediated by PGRMC1, the authors describe some possibilities. The overall manuscript is well written. My comments are described as follows. In the section of 4.2. Environmental regulators of PGRMC1 and PGRMC2, the authors mentioned about regulator as dioxin which induces the PGRMC1 expression. Recent report showed that PPAR agonist increased the PGRMC1 in adipocyte (PMID: 32887925). Several reports suggested that PPAR associated with breast cancer development (PMID: 33087562, PMID: 32850439). These should be discussed. In the section of 6. Biological functions of PGRMC1 and PGRMC2, it is well known that PGRMC1 (and PGRMC2) binds to heme to regulate its activity, and exhibit cancer proliferation and chemoresistance (PMID: 20164297, PMID: 16234411, PMID: 26988023). Is the PGRMC1-mediated P4 action correlated with the heme-biding? These should be discussed. In the section of 7. PGRMCs and the regulation of RNA and protein synthesis, the authors describe the potential for translational regulation from the results of interaction between PGRMC1 and ribosomal proteins, but it does not seem to be supported by the evidence of biological function of PGRMC1. These statements must be simplified or omitted.

Author Response

We thank the reviewers for the insightful comments and kind words regarding our review article.  We have addressed each of the comments as outlined below:

Reviewer #1:

Comment 1:  Recent reports showed that PPAR is associated with breast cancer and that PPAR (PPAR g) increases the expression of PGRMC1. This should be discussed.

Response:  We have addressed this by stating “In addition to dioxins, exposure to environmental pollutants can lead to an increase in PPARg [64].  PPARg is elevated in both ovarian [65] and endometrial [66] cancers and ligand activation of PPARg increases in PGRMC1 in adipocytes [67], making it likely that PPARg is also involved in increasing PGRMC1 levels in these cancers of the reproductive system.

Comment 2:  PGRMC1 and 2 bind heme.  Is P4 actions interaction with PGRMC1 correlated with heme binding?

Response:  This is a good point.  To address this point, we have added the following statements: The capacity to form dimers and oligomers is an important factor in PGRMC1’s action.  PGRMC1 dimers form via an interaction between heme molecules that project from adjacent PGRMC1 molecules [96].  The heme:heme interaction is essential for various aspects of PGRMC1’s actions including the interactions with EGF receptors and cytochromes P450 [96].  Moreover, P4 binds to PGRMC1 at the putative heme/ligand binding cleft and induces conformational change to the heme [96].  How P4 binding influences the configuration of PGRMC1 and therefore its biological function remains to be fully elucidated.

Comment 3:  PGRMC1 interaction with ribosomal proteins are not supported by biological evidence of biological function … and statements [related to this] should be simplified.

Response:  We have strong proteomic data obtained from both ovarian and endometrial cells that PGRMCs interact with ribosomal proteins, consistent with the “ribosomal filter” concept.  The paragraph that presents this ends with the statement “However, this concept remains to be rigorously tested.”  Similarly, the second paragraph on this topic ends with the statement “Whether the abilities of PGRMC1 and PGRMC2 to influence the composition of the ribosomes and thereby which RNAs get translated into specific proteins changes at different stages of ovarian cancer remains to be determined.  These statements make it clear that our proteomic data support but does not prove the ribosomal filter concept.

Reviewer 2 Report

This study was well-written; however some minor mistakes should be revised.

  1. In figure 3,  "E" should be marked in the figure legend.
  2.  In figure 1F, the patient numbers and p value needed to be presented in the figure.

Author Response

We thank the reviewers for the insightful comments and kind words regarding our review article.  We have addressed each of the comments as outlined below:

Reviewer #2:

Comment 1:  Comments are related to lack of information in figures 1 and 3.

Response:  Information has now been added to the legend of figures 1 (patient number and p values provided) and 3 (E is not provided in the legend).

Reviewer 3 Report

The authors performed a very deep review of the progesterone receptor membrane components 1 and 2 and their role in ovarian and endometrial cancers. This manuscript is well planned and written. The references are appropriate. I have no major criticisms for this manuscript.

Minor comments:

  • the authors suggested that the mifepristone mechanism of action is unknown, but it has been shown that mifepristone may act as an agonist through the PGRMC1 receptor;
  • In some places minor editorial errors:

Quantitative PCR measurement confirms that PGRMC1 expression…

Given the expression of these receptors, it is not surprising that P4 plays a role in regulating the growth of ovarian cancers with some studies indicating the P4 stimulates the growth of these cancers

The first published evidence of PGRMC1 expression in the endometrium was demonstrated in a microarray study and confirmed by PCR and Northern blot analyses in which PGRMC1 mRNA expression was found to be substantially higher in…

More recently, orthotoptic orthotopic models have been developed in which tumor cells are transplanted…

Along with the orthotoic orthotopic models, xenograft models using patient-derived…

Author Response

We thank the reviewers for the insightful comments and kind words regarding our review article.  We have addressed each of the comments as outlined below:

Reviewers #3:

Comment 1:  Authors suggested that mifepristone mechanisms is unknown but may act as an agonist to PGRMC1.

Response:  We have included the following to address this comment:  Finally, Mifepristone could actually promote ovarian tumor growth via a PGRMC1-dependent mechanism since the PGRMG1 antagonist, AG 205, has been shown to attenuate Mifepristone growth-promoting actions [5]. 

Comment 2:  Five typos were identified.

Response:  These have been corrected.